# Anatomy of the Male Reproductive System of Sugar Gliders (*Petaurus breviceps*)

**DOI:** 10.3390/ani14182748

**Published:** 2024-09-23

**Authors:** María del Mar Yllera, Diana Alonso-Peñarando, Matilde Lombardero

**Affiliations:** 1Unit of Veterinary Anatomy and Embryology, Department of Anatomy, Animal Production and Clinical Veterinary Sciences, Faculty of Veterinary Sciences, Terra Campus, University of Santiago de Compostela, 27002 Lugo, Spain; mar.yllera@usc.es; 2Department of Animal Pathology, Faculty of Veterinary Sciences, Terra Campus, University of Santiago de Compostela, 27002 Lugo, Spain; diana.alonso.penarando@rai.usc.es; 3Veterinary Clinic Madivet, Calle Comercio, 5, Bargas, 45593 Toledo, Spain

**Keywords:** male genital anatomy, testis, epididymis, deferent duct, penis, corpus cavernosum, corpus spongiosum, bulbospongiosus muscle, ischiocavernosus muscle, prostate, bulbourethral glands

## Abstract

**Simple Summary:**

Sugar gliders are exotic animals that have become pets in recent years. As a result, these small marsupials are increasingly coming to the veterinary practices. Marsupials are a group of mammals characterized by their peculiar way of reproduction. This translates into the distinct anatomy of their genital apparatus, both male and female. There are not many references concerning the anatomy of this particular species. The male reproductive system of the *Petaurus breviceps* follows the general pattern in marsupials, but each species has its own peculiarities that the veterinary clinician should know. It includes two testes, epididymis, and deferent ducts, as well as two types of accessory glands: the prostate and the bulbourethral glands, in addition to the male urethra and the penis, which is bifid at its end. The purpose of this study was to describe the male genital gross anatomy of the sugar gliders (supplemented with some microscopic observations), to provide detailed information to veterinary clinicians to facilitate accurate diagnosis and surgery.

**Abstract:**

The present study provides a detailed macroscopic examination (with some microscopic insights) of the genital apparatus of seven adult and intact male sugar gliders, as well as one castrated individual. The scrotum is pendulous and attached to the ventral abdominal wall, situated in the caudal part of the abdomen and cranial to the cloacal opening. The testes are oval shaped with their long axes oriented vertically. The epididymides are attached along the caudomedial border of their respective testes. The head and tail of the epididymides are in close proximity to the poles of the testes but are not directly attached. The deferent ducts are positioned laterally to their ipsilateral ureter as they run near the dorsal surface of the urinary bladder. The ampulla of the deferent duct is absent. The penis is located post-scrotally, lacks insertion into the bony pelvis, and has a bifid distal end. The crura of the penis originate within the ischiocavernosus muscles, and there are two bulbs of the penis. When flaccid, the entire penis is concealed in the perineal region and externalizes through an orifice situated in the ventral part of the cloaca, traversing its floor towards the cloacal opening. The urethral external orifice is positioned at the point where the bifurcation of the free part of the penis begins. The prostate gland resides in the pelvic cavity and consists of two parts: a macroscopically visible body surrounding the urethra and a microscopically identifiable portion embedded within the walls of the duct. Sugar gliders possess two pairs of bulbous bulbourethral glands, located dorsally and laterally to the rectum, but lack vesicular glands.

## 1. Introduction

The prevalence of novel companion animal species, particularly small marsupials such as sugar gliders, is increasing significantly. These animals now represent a growing segment of the patient population in veterinary practices. Consequently, veterinary professionals are encountering species with anatomical structures, physiological processes, and pathological conditions that diverge substantially from those of traditional domestic animals. However, the paucity of species-specific scientific literature and clinical data compels practitioners to extrapolate information from better-studied species. This extrapolation may potentially lead to complications and inaccuracies in diagnostic procedures, surgical interventions, and therapeutic regimens.

Sugar gliders (*Petaurus breviceps*) are nocturnal, arboreal marsupials indigenous to Australia and New Guinea [1]. These animals exhibit communal roosting behavior, sheltering during daylight hours in tree hollows [2]. In their natural habitat, sugar gliders form social groups characterized by scent-marking behaviors, with the dominant male’s odor serving as a group identifier.

Their social nature makes them a lovable pet that can recognize their owners if they have been properly socialized from an early age [2,3], otherwise, the stress of isolation leads to self-mutilation which, in the case of males, often affects the penis, and amputation of the bifid penis may be necessary if it is severely damaged [4,5]. 

Sugar gliders are omnivorous, with a diet comprising tree sap, nectar, fruits, and insects [4]. Some populations have been observed to include avian eggs and nestlings in their diet [6]. Sexual dimorphism in this species is quite subtle, but males can be distinguished by the presence of scent glands on the forehead and chest [4,5], as well as their pendular scrotum and their larger size.

Marsupials distinguish themselves from other mammals through their unique reproductive processes, which involve distinct anatomical features of their genital apparatus. These differences are particularly pronounced in females, who possess double reproductive organs, but males also exhibit unique characteristics, primarily related to the penis and associated structures.

The physiology, reproduction, and reproductive anatomy and behavior of marsupials have been extensively studied, especially in wild species [7,8,9,10,11,12,13,14,15,16,17,18,19,20,21,22,23,24,25]. However, there is a notable lack of specific references to the internal anatomy of the reproductive system of *Petaurus breviceps*. This study aims to describe the male genital anatomy of the sugar gliders, providing detailed and valuable information to veterinary clinicians to facilitate accurate diagnosis and surgery.

## 2. Materials and Methods

The study sample comprised eight male sugar gliders (*Petaurus breviceps*), including seven intact adults and one neutered individual. The specimens were obtained through owner donations following natural mortality due to various pathologies.

Preservation methods varied among the specimens: four were frozen, one was preserved in formalin, one in 76° alcohol, and two were examined fresh. For one frozen specimen, the entire genital apparatus was extracted post-macroscopic examination and subsequently preserved in formalin for detailed analysis.

Dissections were performed using a Nikon AFX-II magnifying device with magnification ranging from 0.66 to 4.0× and 10× oculars. When necessary, ventral midline incisions were made to access body cavities. Photographic documentation was accomplished using a Coolpix S6 camera (Zoom Nikkor ED 5.8–17.4 mm 1:3.0–5.4) equipped with a macro lens. 

For histological examination, tissue samples were processed using an automatic processor and embedded in paraffin. Sections of 6 μm thickness were cut using a microtome, deparaffinized, hydrated, and stained with hematoxylin-eosin. Following dehydration, slides were mounted in DPX (Sigma-Aldrich, St. Louis, MO, USA). Light microscopy was performed utilizing two distinct microscope setups. The first configuration employed a Nikon microscope coupled with an Olympus EP50 camera (Hamburg, Germany). The second setup utilized a Zeiss Axiophot microscope (Zeiss, Jena, Germany) equipped with a Zeiss AXIOCAM 208 color camera (Zeiss, Jena, Germany). Photomicrographs were acquired at various magnifications using these imaging systems. For low-magnification imaging, a Nikon magnifying loupe was utilized in conjunction with an Olympus EP50 camera.

## 3. Results

The anatomical structure of the male sugar glider’s reproductive tract includes paired gonads, the testes (*testis*), paired gonadal duct systems (each comprises an epididymis (*epididymis*) and a deferent duct (*ductus deferens*)), as well as two accessory glands (the prostate (*prostata*) and a pair of bulbourethral glands (*glandula bulbourethralis*)), the male urethra (*urethra masculina*), and the penis (*penis*).

### 3.1. Testis 

The testes (*testis*) are external in the *Petaurus breviceps*. They are ventral to the caudal part of the abdomen, cranial to the cloacal opening and the penis. Both gonads, together with the ipsilateral epididymis, are enclosed in a cutaneous pouch of skin known as the scrotum (scrotum). The scrotum is fur covered and pendulous, attached to the ventral abdominal wall by a well-developed pedicle measuring 0.5–0.7 cm in length (Figure 1).

The testes are oval-shaped organs with their longitudinal axes oriented vertically. They possess two distinct poles: the head end (*extremitas capitata*) and the tail end (*extremitas caudata*), which are named based on their topographical relationship with the respective parts of the epididymis (head and tail). In the sugar glider, the head end of the testes is positioned dorsally (proximally), while the tail end is situated ventrally and distally. The testes are enveloped by a serous, unpigmented membrane known as the tunica vaginalis (Figure 1). In one specimen, an anomaly was observed, characterized by a prominent longitudinal sulcus on the medial surface of each testis, which was apparent in histological sections but not detectable macroscopically. 

The efferent duct originates from the rete testis and transports spermatozoa towards the head of the epididymis. Histologically, it appears that there is a single efferent duct in the sugar glider. 

### 3.2. Epididymis

The epididymis is a highly convoluted tubular structure that adheres to the caudomedial border of the ipsilateral testis. It is conventionally segmented into three distinct regions: the head, body, and tail. The head (*caput epididymidis*) and tail (*cauda epididymidis*) are situated in close proximity to the proximal and distal poles of the testis, respectively, though they are not directly attached. The body (corpus epididymidis) extends along the caudomedial border of the testis but remains unfused with it. Between the epididymis and testis lies a space known as the testicular bursa (*Bursa testicularis*), which is medially enclosed by the distal mesorchium (*mesorchium distale*) (Figure 2). The tail of the epididymis forms a distinctive rounded protuberance (Figure 1 and Figure 2). Two ligaments are attached to the tail of the epididymis: the proper ligament of the testis (ligamentum testis proprium), which connects it to the gonad, and the ligament of the tail of the epididymis (ligamentum caudae epididymidis), which anchors it to the parietal layer of the peritoneal sac that surround both the epididymis and testicle.

### 3.3. Deferent Duct

The ductus deferens emerges from the tail of the epididymis and runs medially to the body of the epididymis towards the abdomen, enclosed within the testicular spermatic cord (Figure 3a). From the dorsal aspect of the pedicle, the ductus deferens courses subcutaneously in a caudolateral direction until reaching the superficial inguinal ring (*anulus inguinalis superficialis*), to pass through the canal inguinal (*canalis inguinalis*) to enter the abdominal cavity. It is noteworthy that in a minority of cases, anatomical variations may occur. In two out of seven intact male specimens examined, the deferent ducts were observed to cross each other as they entered the pedicle near the head of the epididymis, with the right duct passing to the left side and vice versa. These ducts subsequently recrossed before reaching the abdominal wall, ultimately returning to their original sides.

Upon entering the abdominal cavity, the ductus deferens curves medially to gain the dorsal surface of the urinary bladder neck, running lateral to the ipsilateral ureter (Figure 3b). As the ureters enter the bladder, the ductus deferens shift slightly caudally to enter into the cranial portion of the urethra, passing through the prostate. The ductus deferens maintains a consistent diameter throughout its course. Thus, the ampulla of the deferens duct (*ampulla ductus deferentis*) is absent.

### 3.4. Penis and Urethra

The penis (*penis*) is situated post-scrotally. When flaccid, the entire penis is concealed in the perineal region, extending to the caudal part of the pelvic cavity, with its body positioned ventrally to the rectum, forming an S-shaped curve known as the flexura sigmoidea (*flexura sigmoidea penis*). However, one intact male specimen exhibited a penis whose body extended to the cranial part of the pelvic cavity, with its flexura sigmoidea situated at the same level as the neck of the urinary bladder or the initial part of the urethra. When externalized, the penis emerges through an orifice located in the ventral part of the common cavity known as the cloaca, running along its floor. The cloaca is an internal cavity where the digestive system and urogenital apparatus terminate, communicating with the exterior through a single opening. Upon opening the cloacal orifice to inspect the short cloaca, two holes can be identified: one dorsal where the rectum empties and another ventral through which the bifid end of the penis is visible (Figure 4a). The mucosa of the cloacal floor invaginates to form a sac, known as the preputial sac, where the free part of the penis is located at rest (Figure 4b). At the deepest part of the preputial sac, the mucosa reflects onto the penis, demarcating the boundary between the penile body and the free part (pars libera penis), which comprises approximately half of the total penile length. 

In *Petaurus breviceps*, the free part is bifurcated and caudally oriented (Figure 5). Each bifurcation of the penis exhibits a wider conical morphology near its terminus, with the exception of the castrated specimen. A “urethral groove” extends along the medial aspect of each bifurcation, spanning the entire length to the distal tip. Macroscopic examination reveals a pair of blood vessels visible beneath the penile mucosa, following the course of the aforementioned grooves, one on the dorsal wall and the other on the ventral wall. 

These venous sinuses are discernible both macroscopically and microscopically (Figure 6). Histological analysis confirms the presence of a venous sinus in both the dorsal and ventral walls, corresponding to these observed vessels. The external urethral orifice (*ostium urethrae externum*) is situated in the bifurcation of the penis.

The penis originates in proximity to the caudal surface of the ischial arch (*arcus ischiadicus*). It runs caudally, ventral to the rectum, and terminates at the level of the cloaca. The organ starts with two structures known as the crura of the penis (*crura penis*), the dorsal portion of which is not firmly affixed to the bony pelvis. However, a thin fascia extends from the caudolateral wall of each crus to the ipsilateral ischial arch. The crura, resembling two columns, progress caudoventrally and converge at the midline to form the body of the penis (*corpus penis*). These columns, in conjunction with the two bulbs of the penis (*bulbi penis*), which are situated dorsolaterally to the body, constitute the root of the penis (radix penis).

The penis includes two types of erectile tissue named corpus cavernosum (*corpus cavernosum penis*) and corpus spongiosum (*corpus spongiosum penis*). Both the crura and the bulbs are made of erectile tissue; the crura are made up of the corpus cavernosum while the bulbs are paired, cranial, and lateral expansions of the corpus spongiosum that surrounds the penile urethra. From these bulbs, the erectile tissue extends to form a thin band that encircles the penile urethra. As the urethra transitions into the urethral grooves at the bifurcation of the penis, the corpus cavernosum adopts a “C” shape, positioned laterally to each groove. However, the corpus spongiosum continues to surround the groove, with a particular concentration along its lateral walls. Both types of erectile tissue reach the tip of the penis.

Two pyriform or globose paired muscles are closely associated with the penis of the sugar glider: the bulbospongiosus (*m. bulbospongiosus*) and ischiocavernosus muscles (*m. ischiocavernosus*). The ischiocavernosus muscles are situated caudodorsally to the crura (*crura penis*) and encompass the initial segment of the ipsilateral crus. The bulbospongiosus muscles are positioned dorsolaterally to the body of the penis (*corpus penis*), with one muscle on the right side and the other on the left side (Figure 7 and Figure 8). These muscles are connected to the penis by pedicles that contain extensions of the corpus spongiosum, which surround the penile urethra and are referred to as the bulbs.

The male urethra (*urethra masculina*) extends from the internal orifice at the urinary bladder neck to an external orifice (*ostium urethrae externum*) located at the caudal portion of the corpus, at the point where the bifurcation of the free part of the penis commences. It is composed of two segments: the pelvic part (*pars pelvina*) and the penile part (*pars penina*), which is incorporated within the penis. The entirety of the urethra, both pelvic and penile portions, is enveloped by the corpus spongiosum. 

The body and free part of the penis are covered by a cutaneous mucosa consisting of keratinized epithelium, which also lines the walls of the urethral grooves. However, the bottom of the groove is lined with urothelium. Small cornified spicules may be present at the conical tip of the penis, except in the castrated specimen.

### 3.5. Accessory Glands

#### 3.5.1. Prostate

The prostate of the sugar glider is situated within the pelvic cavity and comprises two distinct parts. First, there is the macroscopically visible body (*corpus prostatae*), which has a carrot-shaped appearance and surrounds the urethra. Second, there is the microscopically observed portion (*pars disseminata prostatae*), located within the walls of the duct. The compacted body envelops the entire pelvic urethra (*pars pelvina urethrae*). Macroscopically, the gland can be divided into three segment (Figure 9): the cranial segment, which is broad and rounded, followed by a constriction that separates it from the second segment, and the third segment, narrower in shape. Following the body of the prostate, the disseminated prostate accompanies the urethra until it is incorporated into the penis, becoming the penile urethra (Figure 9).

#### 3.5.2. Bulbourethral Glands

In the sugar glider, two pairs of bulbous bulbourethral glands (*glandula bulbourethralis*) are observed within the pelvic cavity. These glands are situated dorsally and laterally to the rectum, respectively. They are closely associated with the crura of the penis. Structurally, they exhibit compactness and possess a smooth external surface (Figure 5 and Figure 13). The excretory ducts of these glands run parallel between the bulbospongiosus and ischiocavernosus muscles, ultimately entering the urethra ventromedially just before it transitions into the penile urethra. 

### 3.6. Microstructure

Histologically, the testis is a parenchymatous organ, comprising stroma and parenchyma. The stroma forms a capsule known as the tunica albuginea, which comprises dense irregular connective tissue and encapsulates the parenchyma. The parenchyma is composed of seminiferous tubules interspersed with sparse connective tissue and clusters of interstitial cells (Figure 10). The tunica albuginea is enveloped by a simple squamous epithelium, a serous membrane referred to as the tunica vaginalis lamina visceralis. This, in conjunction with the lamina parietalis, encloses the cavum vaginale.

The absence of septae testis resulted in no observable testicular lobules. The mediastinum testis, along with its rete testis, was situated at the proximal pole of the testis, approximately at a 45º angle from the major axis of the testicle, on the side opposite to the epididymis. Leydig cells, or interstitial cells, were abundant, highly acidophilic, and distributed in richly vascularized clusters. Additionally, a thin layer of myofibroblasts was observed surrounding the seminiferous tubules (Figure 10). 

In relation to the parenchyma, in the seminiferous tubules, the nuclei of sustentacular cells, or Sertoli cells, were discernible in a basal location amidst the male germ cells. These germ cells exhibited a spectrum of developmental stages, from undifferentiated to mature stages, the latter being situated in an apical position, in closer proximity to the lumen. Within a microscopic field, seminiferous tubules exhibited various associations of sperm cell developmental stages. The majority of the parenchyma was constituted by convoluted seminiferous tubules, with straight seminiferous tubules being seldom observed, even in proximity to the mediastinum testis. These straight tubules were lined exclusively by Sertoli cells, or simple to columnar epithelium. 

There seems to be a single efferent duct originating from the rete testis, which courses through the tunica albuginea. This duct, characterized by a mild sinuosity, is lined with an epithelium that varies from squamous-cuboidal to low columnar epithelium. Notably, certain cells within this lining display long apical membrane projections, known as stereocilia.

The epididymis consists of a single, highly convoluted tubule surrounded by loose connective tissue and encased in a dense connective tissue capsule. This tubular structure is arranged in a compact, zig-zag pattern to minimize its overall volume. The epididymis extends from its head (*caput*) through the body (*corpus*) to the tail (*cauda*), where it transitions into the vas deferens. When examined in longitudinal section, the microstructure of the epididymis reveals distinct lobuli epididymidis separated by thin transverse septa composed of dense connective tissue. Up to 15 lobuli epididymidis have been observed, with the largest and most bulky being the one from the tail (Figure 10). The ductus of the epididymis is lined by a pseudostratified columnar epithelium that displays varying morphology (Figure 10), with its thickness progressively increasing from the head towards the caudal portion. Based on these morphological variations in the pseudostratified epithelium, the epididymal duct can be divided into five distinct zones, each with its unique cellular characteristics. Similarly, the smooth muscle surrounding the ductus in a circular layer also increases in thickness from the head to the tail. This muscular layer is notably thin in the caput and progressively thickens to several cell layers thick in the tail, as a transition to the deferent duct. 

Upon integration into the spermatic cord, the deferent duct lumen is significantly wider than that of the epididymis duct (Figure 10) and is lined with cuboidal to low columnar epithelium. The muscular layer is substantially thickened, appearing to consist exclusively of circularly oriented smooth muscle cells. It is accompanied by numerous sinuous arterial vessels and the veins of the pampiniform plexus, and the cremaster muscle.

Both vasa deferentia terminate at the roof of the prostatic urethra, forming a small elevation known as the seminal colliculus. It seems that there is no crista urethralis between the termination points of the ducts. The point of confluence of the vasa deferentia with the prostatic urethra is located in the cranial region of the prostate, but approaching the middle portion of the gland.

The penis is anchored by two lateral projections of dense connective tissue, which are introduced as an oblique sheet into the ischiocavernosus muscle, apparently without fixation to the ischial arch. These lamellar structures of dense connective tissue thicken, and blood vessels arrive at them that are incorporated into their inner part. In this way, these structures widen and gain volume in their medial part and incorporate the pelvic urethra, becoming penile urethra. This remains in a ventral position and the connective tissue structures merge in the midline and present large spaces filled with blood (caverns), forming the corpus cavernosum, in a dorsal position to the urethra. Thus, the penis is visualized as formed by two columns or longitudinal structures of dense connective tissue (of semilunar section) fused in their dorsal midline by a shared septum of dense connective tissue. Inside they present septa or trabeculae of dense connective tissue with radial arrangement from the urethra to the periphery and vice versa, which can be complete (when they cross the entire cavity) or incomplete and among which appear spaces filled with blood. They would form the corpus cavernosum of the penis (Figure 11). 

At the level of the body of the penis, the urethra is a closed duct, lined by urothelium, which conducts urine and semen. In the proximal part of the penile urethra, the ducts of the bulbourethral glands join, potentially contributing more constituents to form the semen. The penile urethra runs in a ventral position to the corpus cavernosum and in its course it is surrounded by a multitude of blood vessels between its muscular wall, which form the corpus spongiosus. The penile urethra remains closed until the formation of a ventral groove that reaches it and opens it; this point is considered the external urethral orifice, just before the bifurcation of the free part of the penis (the part that is externalized during mating) (Figure 11). This groove becomes deeper and separates the two cavernous bodies, leading to the bifurcation of the penis. However, there is still an area that continues to be lined by transitional epithelium that transports semen to the tip of each part of the bifurcated penis. At this point, the penis begins to be externally covered by cutaneous mucosa all around its perimeter, originating the free parts of the penis, free of unions or adhesions to the prepuce. In the bifurcated free parts, a deep groove appears whose bottom is lined by urothelium, forming the urethral grooves. On both sides of the walls of the groove there are venous sinuses visible macroscopically and microscopically. And the cavernous body forms a concave or grooved structure, whose concave part protects the urothelium of the urethral groove, in a medial position, looking towards it contralateral. 

At the terminal ends of each bifurcation of the free part of the penis, there is a conical widening formed by the corpus cavernosum that partially surrounds the urethral groove. This urethral groove does not end exactly at the vertex of the conical structure, but very close to it. Histologically, this conical dilation is formed by the cavernous body that surrounds the urethral groove, under whose urothelium the spongy body is arranged. This structure appears externally covered by cutaneous mucosa that presents small cornified spicules, without apparent orientation (Figure 11).

The prostate is a parenchymatous gland enveloped by a dense connective tissue capsule. Histologically, it can be divided into three distinct segments based on the glandular tissue types: the cranial (anterior) segment, the middle (central) segment, and the caudal (posterior) segment (Figure 12). Within this glandular tissue, simple branched tubular glands exhibit a radial arrangement. Their basal closed ends are in contact with the connective capsule, while their openings converge toward the urethra. These openings do not directly connect to the urethra but instead lead to single secretory ducts lined by cuboidal-columnar to transitional epithelium, which ultimately release their secretions into the urethra. Each glandular tubule is enveloped by a thin layer of smooth muscle. In the three sections of the prostate, the tubulo-alveolar glands have a different appearance (Figure 12). In the cranial portion, the glandular epithelium that covers the outermost region is formed by columnar cells, while towards the part closest to the urethra, the cells are more cubic. Inside the lumen appears acidophilic secretion in the form of individualized droplets of different diameters. Both in the middle and caudal parts, no differences are observed between the outer and inner parts. The glands in the middle part are internally lined by small basophilic cubic cells that release a basophilic secretion. In contrast, in the caudal part, the glandular epithelium is made up of large cells with clear cytoplasm that release a secretion forming large independent droplets of colorless appearance.

The prostatic urethra receives the prostate secretion via numerous ducts and is encircled by erectile tissue containing small vessels that contribute to the formation of the corpus spongiosum (Figure 12). Toward the end of the prostate body, glandular tissues become visible within the thickness of the urethral walls, situated beneath the smooth muscle layer. This arrangement constitutes the disseminated prostate. In the final segments of the pelvic urethra, the glandular epithelium emerges from the muscular layer and is positioned around the urethra.

The two bulbourethral glands of the sugar glider have different histological characteristics, as well as the appearance of their secretion. Both are tubulo-alveolar exocrine glands surrounded by a capsule of dense connective tissue that emits septa of loose connective tissue among which the glandular epithelium is arranged, forming a single layer of cells that pour their secretion towards the light (Figure 13). In bulbourethral gland I (which is located closer to the pillars of the penis), the glandular cells are cubic, with clear cytoplasm, while in bulbourethral gland II, the secretory cells are low cubic and produce a very acidophilic secretion that is stored in the light of the alveoli, with granules of more intense staining. Both glands are externally surrounded by a double layer of striated muscle tissue, oriented perpendicular to each other. Their ducts convey their secretion into the penile urethra (Figure 13).

## 4. Discussion

The male reproductive system of the sugar glider is representative of marsupials and exhibits a greater similarity to the male genitalia of placental mammals as compared to the female counterpart [26]. This system fundamentally comprises the testes, which is the site of gamete formation, a duct system that transports the spermatozoa, and a collection of glands that are responsible for the secretion of seminal fluid [7]. Previous investigations into this system have been undertaken by Ness and Booth (2004) [27], Carboni and Tuly (2009) [4], and Girling (2013) [28].

The results obtained from our observations, at both macroscopic and microscopic levels, will be collectively discussed.

### 4.1. Testes

As in other marsupials, the scrotum of the sugar glider, with the testes and the epididymis inside, is always external and cranial to the cloacal opening [4], as well as to the penis [5,8,15,16,21,29,30]. Its pendular nature, characterized by a lengthy scrotal neck [27], follows the general pattern in marsupials, facilitating the cooling of the testes and epididymis by radiation and, in some species, also via the evaporation of sweat or saliva [7]. The cremaster muscle, which extends along the scrotal stalk, further aids in the thermoregulation of the testicle, drawing it nearer to the body in cold climates or distancing it in warmer conditions [7]. Only a few marsupial burrowing species, such as the *Notoryctes* genre (marsupial moles), lack this pendulous scrotum. Their testes are positioned pre-penially, but they lie between the skin and the abdominal wall [7], just cranial to the pubic bones [31,32]. In the southern hairy-nosed wombat *(Lasiorhinus latifrons)* and the common wombat *(Vombatus ursinus)* [15,16], the scrotum is situated close to the body and is non-pendulous. A similar scenario is observed in the *Acrobates pygmaeus*, the smallest Australian glider [14].

Based on our findings, the testes of the *Petaurus breviceps* appear to conform the general marsupial norm; they possess an ellipsoid shape and they are permanently located within the scrotum [22]. Contrary to our observations in the sugar glider, the tunica vaginalis that encapsulates the testes appears to be to be melanized in numerous marsupials [8,12,33]. Nevertheless, the sugar glider is not the sole species where it is not of a dark hue. This has also been documented in the *Acrobates pygmaeus* [14], the common wombat (*Vombatus ursinus*) [15], the hairy-nosed wombat *(Lasiorhinus latifrons)* [16], and certain South American marsupials: Caenolestes [12], *Didelphis sp.* [30], and *Chironectes minimus* [23]. The functional or phylogenetic importance of this characteristic remains undetermined. Biggers (1966) [34] proposed a potential role in the regulation of testicular temperature, yet there appears to be no correlation between climate and tunic pigmentation as might be expected [22]. However, the findings of de Barros et al. (2013) [30] seem to emphasize this hypothesis, giving that the young and sexually inactive Didelphis sp. exhibited a whitish tunica vaginalis, while the adults displayed a high degree of dark pigmentation.

### 4.2. Epididymis

The epididymis is similar in shape to that of eutherians and other marsupials. However, in numerous marsupials, the head of the epididymis is not affixed to the testicular capsule as we observed in the *Petaurus breviceps*, but it is connected by the efferent ducts that transport the sperm from the testicle [35,36]. Contrary to placental mammals, a singular efferent duct has been documented in many marsupials, encompassing both Australian and South American species [8,11,35], albeit not universally [37]. Histologically, in our studied specimens, it appears that several straight ducts (lined with pale low-cuboidal epithelium) merged into a unique efferent duct that departs from the testis and reaches the head of the epididymis. The duct is lined by a dark, low ciliated columnar epithelium. Moreover, these ducts do not seem to follow a straight course, but rather appear coiled and are surrounded by a plexus of small blood vessels. Despite the requirement for additional histological studies to validate the distinct sections of the duct system between the testis and epididymis, certain parallels were observed with the descriptions provided by Woolley (1987) [38] in American marsupials. However, none of the descriptions corresponding to the various species entirely aligned with our findings in the sugar gliders.

In general, the position of the epididymis in relation to the testicle is not explicitly defined in the literature. Only de Barros et al. (2013) [30] have described it as lateral to the testis in certain Brazilian species *(Didelphis sp).* In contrast, the position of the epididymis in the sugar glider is caudomedial, similar to the arrangement found in domestic mammals with a pendulous scrotum [39].

The deferent duct establishes a connection between the epididymis and the prostatic portion of the urethra [39,40]. However, due to the migration of the ureters during embryonic development of the urogenital tract, the relative position of the ureter and the deferent duct varies between different groups of mammals. Specifically, in marsupials, the ureters are medial to the genital ducts (mesonephric duct in males), while in eutherians, they are lateral [41]. Consequently, in *Petaurus breviceps*, as is the case with other marsupials, the deferent duct is positioned laterally to the ipsilateral ureter.

### 4.3. Deferent Duct

In the majority of eutherian species, the final segment of the deferent duct, which lies on the bladder, displays a fusiform enlargement. This is known as the ampulla of the deferent duct, where the ampullary glands are located [39]. In contrast, the sugar glider, like other marsupials, maintains a consistent thickness throughout the length of the deferent duct [15]. Consequently, due to the absence of the ampulla of the deferent duct, ampullary glands are also absent [4,40]. However, exceptions may exist. For instance, a convoluted structure resembling an ampulla has been described in *Caenolestes obscurus* by Rodger in 1982 [12]. Histologically, we have observed that both vas deferens reach the roof of the prostatic urethra forming a small elevation known as the seminal colliculus. However, this elevation is not as pronounced as in domestic placental mammal species. The point of confluence of the vas deferens into the urethra is in the cranial part of the prostate but in its posterior part, very close to the middle part of the prostate. Our results contrast with those reported by Ward and Renfree (1988) [14] in *Acrobates pygmeus,* another Australian glider. They found that both vas deferens empty into the urethra at the level of the middle or central segment of the prostate.

### 4.4. Penis 

The penis of the *Petaurus breviceps* is situated in a post-scrotal position, a characteristic shared by both South American and Australian marsupials [10,15,23,28]. 

Existing literature indicates that, during rest, the penis of the sugar glider is located inside the cloaca, specifically lying on its ventral wall [4,28,29,40]. However, our detailed dissections, revealed a different arrangement. The cloaca of this species is notably short, and the penis is actually positioned ventral to the rectum. During copulation, the free part of the penis is externalized through an orifice located in the caudoventral wall of the cavity. This anatomical configuration aligns with descriptions provided by Nogueira et al. in Brazilian didelphids [10] and the koala [25]. The cloaca, common to urogenital apparatus and the digestive system, is a common feature in marsupials. An exception to this is *Chironectes minimus*, where the digestive tract has its own separate opening, distinct from that of the urogenital tract [23].

The sigmoid flexure (*flexura sigmoidea penis)* that we saw in our specimens was previously described by other authors [22,28], and seems to be a consistent characteristic among marsupials [10,15,19,23,24,25].

According to the *Nomina Anatomica Veterinaria* (N.A.V.) [42], when at rest, the end of the penis is concealed by a fold of abdominal skin known as the prepuce *(preputium).* This fold’s internal layer is attached to the organ. In the sugar glider, the penis, when flaccid, is hidden within the body, positioned ventral to the rectum, rendering the prepuce absent. However, the term ‘prepuce’ is often used in the literature to refer to the fold of the cloacal mucosa that attaches to the marsupial penis. Similarly, the sac housing the resting penis is called the preputial sac [10,15,19,25]. Precisely, in placental mammals, the attachment of this skinfold demarcates the separation between the body and the free part of the penis. In *Petaurus breviceps*, this boundary could potentially be identified by the location of the external urethral orifice and the bifurcation of the terminal part of the organ. However, as the urethra does not extend to the end of the penis, the structure of the eutherian free part of the penis cannot be recognized. Moreover, it is also not possible to identify the typical mammalian glans *(glans penis)*, which, according to the N.A.V. [42] is characterized by its position at the end of the free part and being formed exclusively by the urethra surrounded by some erectile tissue, the corpus spongiosum *(corpus spongiosum glandis).* Despite this, the terms ‘free part’ and ‘glans’ of the penis are frequently used in the referenced literature [7,22,24], even for species with a forked penis [10,15,17,21,25,30,43,44]. This fact could be attributed to the consideration of the glans penis as the terminal portion of the penis, where the corpus cavernosum expands and surrounds the corpus spongiosum, as suggested by Woolley and Webb (1977) [25].

In summary, knowing the distribution of the erectile tissue at the end of the penis is crucial to confirm the presence of a ‘proper’ glans penis in a species. To our knowledge, it is absent in most marsupials, as demonstrated by Warburton et al. (2019) [19] in their study on the red kangaroo. Histological studies conducted on Australian and New Guinea dasyurids, which possess slight or non-bifid penises [9,13,25,45] revealed that the corpus cavernosum consistently extends to the tip of the penis. Similar results have been obtained in studies on South American marsupials with bifurcated penises; the two types of erectile tissue extend to the tip of the penis [10,30,46]. This is a characteristic we also observed in the *Petaurus breviceps*. Other histological studies conducted on dasyuroides with a non-bifid penis by Woolley (1987) [45] revealed the presence of only a corpus cavernosum at the tip of the penis. Consequently, the existence of a true glans penis, as defined by the N.A.V. [42], cannot be confirmed in marsupials.

The distribution of the corpus cavernosum in the bifurcated free part of the penis in the bushy-tailed opossum (*Glironia vetusta*) [46] is similar to our observations in the sugar glider. In both instances, this erectile tissue forms two semilunar structures situated laterally to the urethral grooves. However, the arrangement of the corpus spongiosum differs; it surrounds the cavernosum in the bushy-tailed opossum, as it does in other Australian and South American marsupials, both with slightly or non-bifid free part of the penis [9,10,13,17,25,46], while in the sugar glider, it is positioned around the urethral groove.

In certain species with a forked penis, a dorsal projection known as the dorsal lobe, containing a corpus spongiosum, is present at the beginning of the free part [17]. This feature was not observed in *Petaurus breviceps*. Additionally, we were unable to identify the accessory corpora cavernosa, which forms an appendage of the penis as described by Woolley and Webb [25], and Woolley [17] in some Australian and New Guinean dasyurid marsupials with a non-bifid penis tip.

The bifid tip of the penis is a common feature in marsupials, as described in certain species of *Sminthopsis* [13], *Antechinus* [17], Brazilian *Didelphidae* [10], and the common wombat *(Vombatus ursinus)* [15]. However, it is not observed in *Dasyuroides byrnei, Dasycercus cristicauda* [45], *Dasykaluta, Pseudantechinus, Parantechinus* [17], or the grey kangaroo *(Macropus fuliginosus)* [19]. The bifurcated free part of the penis varies between species in terms of length, shape of the tip, the presence or absence of a diverticulum, and the positioning of the urethral grooves. In the sugar glider, the absence of a diverticulum, the considerable length of the free part, and the urethral grooves—located in the medial wall end very close to the tip of each half—are similar to the features observed in *Caluromys,* a South American marsupial. In contrast, other didelphidae *(Didelphis, Lutreolina, and Metachirus)* have a shorter free part with a diverticulum, and the grooves, also medial, do not extend to the end of the penis [10]. Among Australian marsupials, *Antechinus* species also exhibit long urethral grooves [17], akin to the sugar glider.

For all species that have a bifurcated penis [10], the urethra terminates at the caudal part of the body of the penis, between the body of the penis and the free part, as described in the literature for the sugar glider [3,5] and in line with our own observations. This specific anatomical feature is of considerable relevance, given that males do not urinate from the forked end of the penis, but from the proximal portion [27]. Consequently, the distal penis can be amputated in cases of penile necrosis, penile trauma, or paraphimosis. When an amputation procedure of the distal penis is required [47], the amputation does not compromise the survival of the individual. This is attributable to the fact that the urethra does not pass through the bifurcated sections of the penis but extends as a urethral groove along the medial surface of each half. This implies that even in the event of an amputation of one or both of the free, bifurcated parts of the penis, the male retains the ability to urinate without any complications, as the urethra remains intact and fully functional. Given that penile self-mutilation is relatively common in stressed captive males, the position of the urethral orifice enables the amputation of the terminal ends, when necessary, without causing damage or alterations to the urinary tract [3,27]. 

The same situation of the urethral opening has been previously described in the *Antechinus stuartii* [25]. The urethra opens into urethral grooves, the location of which appears to be species-dependent. Woolley et al. (2007) [13] conducted a morphological study of the penis in *Sminthopis* species (dasyuridae). They discovered that in species with a slightly bifurcated tip, the external urethral orifice is located dorsally on the free part of the penis, next to its end. The urethral grooves are either on the mesial aspect (in twelve species) or on the ventral aspect (in one species) of the bifid portion of the penis. However, in a previous study carried out in *Dasyuroides byrnei* and *Dasycercus cristicauda* (Dasyuridae), the same author [38] described the urethral grooves in the dorsal part of the penis tip. In *Antechinus*, six out of seven species with a forked penis exhibited the urethral grooves on the dorsal surface; in the seventh species, they were dorsally located in the cranial part of the free portion but they were present on the medial surface of the apices [17]. The author proposed that this positional change could be attributed to the curving of the apices towards the midline. In *Petaurus breviceps,* Carboni and Tully (2009) [4] and Johnson-Delaney and Lennox (2017) [48] identified a ventral groove in each part of the forked end of the penis. Our observations revealed a similar structure, but it was medially (mesially) positioned when the organ was placed in the physiological position, as seen in the Virginia opossum *(Didelphis virginiana)* [43], Brazilian *Didelphis sp.* [30], *Chironectes minimus* [23], and *Antechinus stuartii* [25]. The length of these urethral grooves varies among species: in the sugar glider, they extend almost to the tip of each part, similar to Chironectes minimus *Chironectes minimus* [23] and *Caluromys* [43], whereas in *Didelphis virginiana,* they terminate some distance from the tips [43].

The *Nomina Anatomica Veterinaria* (N.A.V.) [42] describes three parts in the penis of placental mammals: the root *(radix penis)*, the body *(corpus penis),* and the free part *(pars libera penis).* The root comprises two crura (*crura penis*) and the bulb of the penis *(bulbus penis*). Each crus is essentially the proximal part of a corpus cavernosum anchored to the *archus ischiaticus* and enveloped by the ischiocavernosus muscle. The bulb of the penis represents the caudal expansion of the corpora spongiosum penis and is sheathed by the bulbospongiosus muscle [42].

In placental/domestic mammals, the corpus spongiosum *(corpus spongiosum penis)* commences at the pelvic outlet with a bilobed expansion, constituting the bulb of the penis *(bulbus penis).* Subsequently, it narrows and encapsulates the segment of the urethra situated within the penis [39]. A distinctive feature of placental mammals is the presence of a singular penile bulb, contrasting with marsupials that exhibit dual expansions in the cranial region of the corpus spongiosum, situated on the right and left sides [9,10,13,19,21,25,30]. Regardless of the species, the penile bulb(s) are enveloped by the bulbospongiosus muscles.

The bulbospongiosi muscles, due to their form and location on either side of the penile body, and the slender, duct-like pedicles connecting them to the penile corpus, can be easily mistaken for bulbourethral glands, as noted by Young in his study of the koala [21].

The anatomical configuration and globular shape of the *bulbospongiosi* and *ischiocavernosus* muscles as observed in the *Petaurus breviceps,* have been previously documented in both South American and Australian marsupials [10,19,21]. In these species, the right and left *bulbospongiosi* muscles remain distinct from each other, unlike in placental mammals where they merge ventrally to form a single bulb. The significance of this anatomical difference remains ambiguous. Various hypotheses have been proposed, but none have been definitively validated. Young postulated in 1879 [21] that the double bulb might be essential to sustain the turgescence of the bifurcated glans, a characteristic feature of most marsupials. However, given that the corpus spongiosum does not bifurcate within the penile body, Warburton et al. (2019) [19] deem any functional differentiation between the right and left muscles improbable. According to these authors, the observed characteristics likely represent the bilateral development of the reproductive organs.

However, the anatomical differences between the bulbospongiosus and ischiocavernosus muscles in placental mammals and marsupials imply divergent functions. In placental mammals, the ischiocavernosus muscles contribute to the augmentation of blood pressure in the corpus cavernosus of the penis, facilitating erection [49]. The rhythmic contractions of the bulbospongiosus muscle aid in the propulsion of sperm through the urethra during ejaculation [39,43,50], enhance the erections of the glans penis [43,50], and the contractions of the ischiocavernosus muscle elevate the penis, assisting the male in achieving the intromission [43]. These contractions also induce or assist in penile movements, such as the “penile flips” observed in rats [50]. In marsupials, according to Warburton et al. (2019) [19] and Dixon (2021) [43], the function of both muscles could be construed as “mechanical pumps” that increase the pressure in the erectile tissues during the erection and ejaculation, However, these muscles are not believed to play a role in the eversion of the penis from the cloaca or in the execution of penile movements.

In our study, we identified the presence of horny or epidermal papillae, also referred to as spines, at the tip of the penis. This observation aligns with descriptions of the free portion of the penis in other marsupial species, both Australian [13,15,16,21,45,51] and American [10,30]. However, in this small species, given the size of these structures, we prefer to call them spicules, rather than spines. The morphology and arrangement of these penile spines vary significantly across species, with different shape and orientations. For instance, in the koala they are retroverted [51], whereas in the sugar glider they do not appear to possess a specific orientation. The function of these spicules in marsupials remains undetermined. In certain placental mammals, analogous structures are associated with the necessary vaginal stimulation for ovulation, facilitated by the release of hormones in ovulation-induced females. However, existing studies in marsupials have yet to yield definitive results [52]. Additional proposed functions of the spicules include inducing physical alterations in the female genital mucosa to facilitate the penetration of seminal substances responsible for triggering ovulation, providing tactile feedback to the male during copulation, or potentially playing a role in sexual selection [53]. However, none of these studies include marsupial species. One specimen examined in our study was a neutered male, which exhibited an absence of epidermal spicules at the tip of the penis. While no histological analysis was conducted to confirm their absence, no spicules were detectable under binocular loupe examination. This may be attributable to the fact that, as in other mammals, castration induces the regression of penile spines, given that they are secondary sexual characteristics and thus testosterone-dependent, as extensively documented in the male cat [54,55].

### 4.5. Accessory Glands

#### 4.5.1. Prostate

According to the available literature, the prostate of the *Petaurus breviceps* is a large and disseminate gland [4,28,40,48]. The *Nomina Anatomica Veterinaria* [42] characterizes the prostate as a glandular structure composed of two distinct parts: a compact component termed as the “body” *(corpus prostatae)*, and a diffusely distributed part within the wall of the pelvic urethra *(pars disseminata prostatae).* Given that the authors identify a large, macroscopic prostatic portion enveloping the urethra, the prostate of the sugar glider does not align with the disseminated part and should be referred to as the body. Furthermore, it is a compact structure. In terms of its anatomical structure, it comprises three macroscopically distinguishable regions: cranial, media, and caudal, referred as anterior, central, and posterior segments in the literature. However, the *Nomina Anatomica Veterinaria* [56] stipulates that the terms *‘anterior’ and ‘posterior’ (as well as ‘superior’ and ‘inferior’)* ‘cannot be generally applied to quadrupeds because of the confusion arising from their meaning in human anatomy’. Consequently, the application of these terms is confined to certain structures of the head, excluding other parts of the body. Therefore, in the context of the sugar glider’s prostate, being a quadruped animal, the part contiguous to the urinary bladder should be designated as the cranial part, and the part oriented towards the tail should be termed as the caudal part. The part situated between these two extremes could be interchangeably referred to as the central part or media.

In the sugar glider the prostate displays a morphology that is analogous to the shape of a carrot, a form that is predominantly observed among marsupials [15,16,23,24,30,57]. The constriction that segregates the initial two portions has been previously described in *Petaurus breviceps* by Johnson-Delaney (2021) [5]. A heart-shaped prostate has been reported in a few species, such as the *Acrobates pygmaeus* [14], bandicoots *(Isoodon macrourus* and *Parameles nasuta)* [24], and the koala *(Phascolarctos cinereus)* [21].

Histological studies of the prostate of marsupials have revealed that it is divided into several regions, each comprising different types of glands that produce a variety of secretion products, with some variation observed between family groups [20,24]. In the sugar glider prostate, we observed a transverse division into three zones: cranial (or anterior), media (or central), and caudal or posterior. To our knowledge, this tripartite transverse segmentation is common in marsupials [11,14,15,16,23,24,30,57]. However, despite its carrot-like shape, the distribution pattern of the three parts may vary depending on the species [24]; the limit of the cranial part is always transverse, but the former central and the caudal part could exhibit a different distribution, hence the nomenclature should be altered to internal/external depending on their distribution. Additionally, several exceptions have been documented with only two morphological and functional zones in the prostate gland: dorsal and ventral in the bandicoot *(Isoodon macrourus)* [24,58] or anterior and posterior in the honey possum (*Tarsipes rostratus)* [59].

#### 4.5.2. Bulbourethral Glands

In the majority of placental mammal species, males possess a pair of bulbourethral glands, also known as Cowper’s glands. These are situated adjacent to the rectum and discharge into the caudal section of the pelvic urethra. The literature reveals a variation in the number of bulbourethral gland pairs in marsupials, with some sources citing one, two, or three pairs [15,18,24,28,40]. According to our study, the sugar glider has two pairs, which aligns with the existing bibliography [4,5,40,48]. These glands positioned dorsally and laterally to the rectum, were initially characterized by Carboni and Tully (2009) [4] as multilobed. Other authors refer to them as lobulated as well [7,40]. However, our observations suggest a smooth external surface and a compact structure. Rodger and Huges (1973) [24] noted interspecies variation in gland shape, and even intraspecies variation between different gland pairs. The glands may exhibit a kidney or bulbous shape and may or may not be lobed. This observation is consistent with the findings of de Barros et al. (2013) [30] in Didelphis sp. Furthermore, these authors described three pairs of bulbourethral glands, each with a distinct glandular epithelium, but all encased within a double layer of striated muscle. In the same way, histological differences in the epithelium of the three gland pairs have also been reported in the hairy-nosed wombat (*Lasiorhinus latifrons*) [16] and the *Didelphis virginiana* [11].

In species that possess two pairs of bulbourethral glands, these can be situated dorsally to the pelvic urethra, as observed in the sugar glider, or alternatively, one pair may be dorsal and the other ventral, as in the case of *Chinorectes minimus* [23]. In instances where a third pair is present, two pairs are positioned dorsally and one ventrally to the urethra. This configuration is evident in the hairy-nosed wombat *(Lasiorhinus latifrons)* [16] and *Didelphis sp* [30,60]. The ducts of these glands discharge into the proximal part of the penile urethra [4], in close proximity to the crura penis. Each gland’s duct enters the urethra on the ventral surface near the crus penis [40,48]. In the case of *Didelphis albiventris*, three pairs of bulbourethral glands are present: lateral, intermediate, and medial. The ducts of the first two pairs converge near the urethra to form a common duct, which, in conjunction with the duct of the medial gland, leads to a conical papilla ventrolateral to the initial section of the penile urethra [60]. However, in the sugar glider, our observations indicate that both ducts join the urethra in a caudal position, just cranial to the crura penis. The ducts initially run dorsally to the urethra before incorporating into the urethra, where they continue laterally, remaining independent and opening further cranially, without a papilla. To the best of our knowledge, this specific course of the bulbourethral ducts has not been previously documented in marsupial literature.

Although the histology of the bulbourethral glands of *Petaurus breviceps* has not been previously studied, the striated muscle envelope surrounding each gland has been previously described in other species [11,16,24,30,57].

According to previous studies, marsupials lack vesicular and coagulating glands [4,12,15,23,24,28,30,40,48,61,62]. Our findings corroborate the conclusions of these earlier investigations.

## 5. Conclusions

Sugar gliders (*Petaurus breviceps*) are becoming increasingly popular as marsupial pets due to their diminutive size and sociable disposition, which facilitates interaction with their owners. Consequently, these animals are more frequently presented as patients in veterinary clinics. However, veterinarians often face challenges in obtaining comprehensive information about their anatomical structure. Our study provides insight into the male reproductive system of *Petaurus breviceps*, that consists of two testes, epididymides, and deferent ducts, a single prostate, two bulbourethral glands, and a bifurcated, post-scrotal penis, a feature commonly misinterpreted as abnormal by pet owners. While not prevalent, certain pathologies such as penile fibrosarcoma and prostatitis have been documented in marsupials, including sugar gliders. In contrast, penile mutilation and necrosis are commonly observed in pet *Petaurus breviceps*. These conditions are typically stress-induced, often due to inadequate care in captivity, such as when these animals are kept in isolation or with an aggressive cage mate. In certain instances, surgery may be required to remove the damaged part or even the entire bifid portion of the penis. In such cases, a thorough understanding of the male genital tract’s anatomy is essential for accurately diagnosing and effectively treating the pathological condition.

## Figures and Tables

**Figure 1 animals-14-02748-f001:**
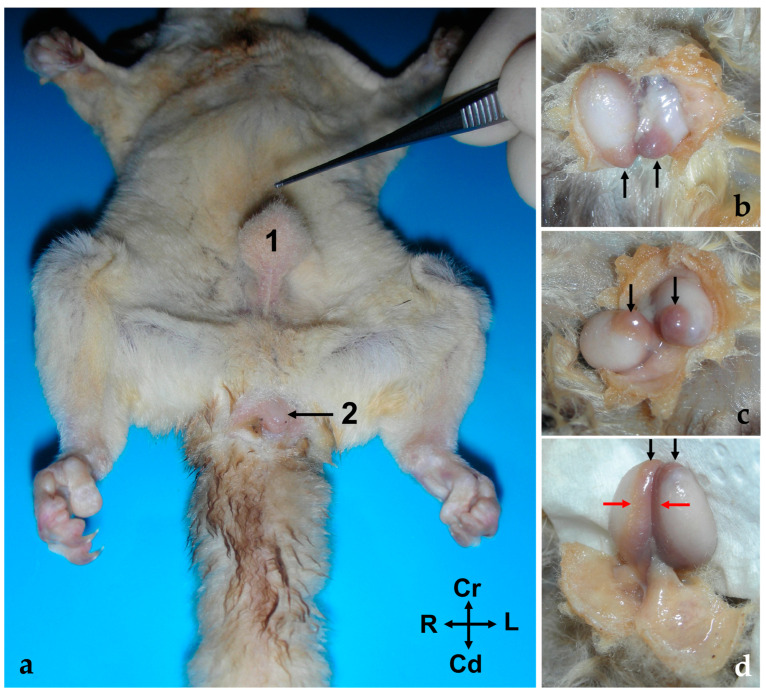
(**a**) Ventral view of the abdominal wall of a male sugar glider (fresh specimen); 1, the scrotum; 2, the cloacal orifice. (**b**–**d**) Images providing a detailed view of the testes and epididymis in their natural position. The scrotum has been incised for better visibility. The black arrows indicate the tail of the epididymis, while the red arrows point to the body of the epididymis. (**b**) Cranial view, (**c**) ventral view, and (**d**) caudal view. All photographs maintain the same orientation as indicated in image (**a**).

**Figure 2 animals-14-02748-f002:**
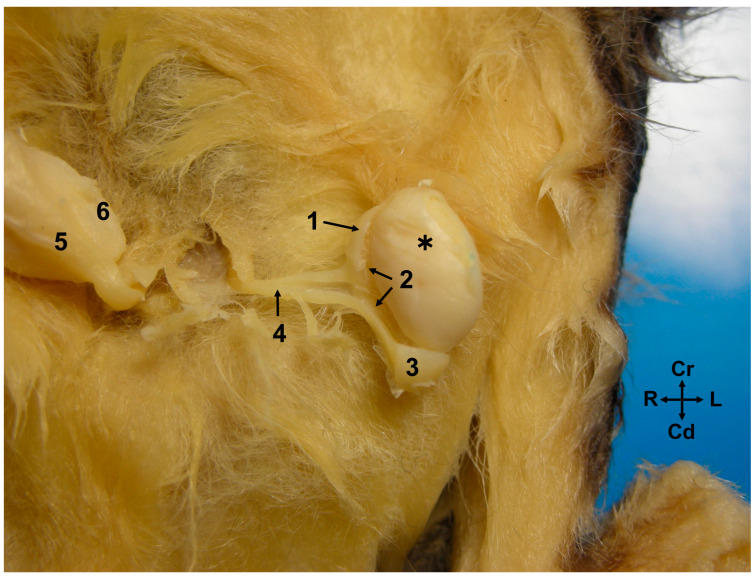
Ventral view of the abdominal wall of a male sugar glider. The *scrotum* has been incised (alcohol-fixed specimen). The left testis (**✱**) is in a displaced position, and the left tunica vaginalis parietalis has been excised. Left epididymis: 1, head; 2, bursa testicularis, a space situated between the body of the epididymis and the testis; 3, tail; 4, left deferent duct; 5, right epididymis; and 6, right testis.

**Figure 3 animals-14-02748-f003:**
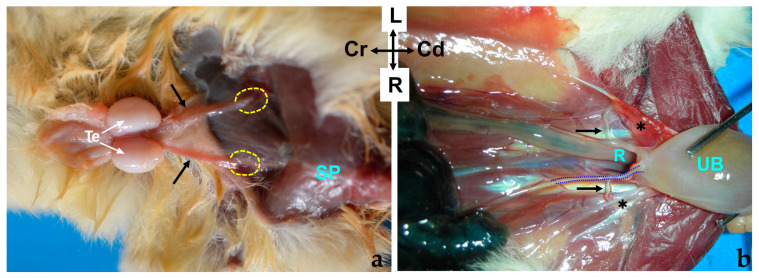
(**a**) Ventral view of a male sugar glider’s abdominal wall. The extra-abdominal course of the deferent ducts is indicated by black arrows, leading towards the superficial inguinal ring, represented by a discontinuous oval shape; Te, testes (**b**) ventral view of a male sugar glider’s abdominal cavity, with the digestive viscera displaced cranially. The intra-abdominal trajectory of the deferent ducts, again indicated by black arrows, leads towards the dorsal and cranial prostate. The urinary bladder (UB) has been reflected in a caudal direction to reveal its dorsal surface and the beginning of the prostate gland. The right deferent duct is demarcated by a red dotted line, while the terminal segment of the right ureter is indicated by a blue dotted line. Both males were fresh specimens. R, Rectum; SP, symphysis pubica; Te, testes; UB, urinary bladder; ✱, lateral vesical ligaments with round ligaments of the bladder. The orientation is the same in both images.

**Figure 4 animals-14-02748-f004:**
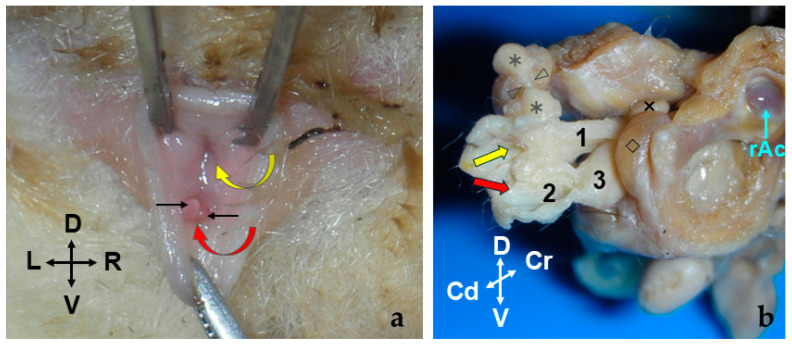
(**a**) Caudal view of a male sugar glider. The cloacal orifice is opened to reveal the opening of the rectum (yellow arrow) and the tips of the bifurcated penis (red arrow) positioned in the roof and floor of the cloaca, respectively. The forked end of the penis is visible protruding through the hole in the floor of the cloaca (black arrows). Fresh specimen. (**b**) Right lateral view of the caudal part of a male sugar glider. Dissection reveals the terminal sections of the digestive and urogenital systems, with the right cloacal wall removed. 1, Rectum; 2, preputial sac; 3, penis inside the preputial sac; ◇ right bulbospongiosus muscle; ✕, right bulbourethral gland; ✱, paracloacal glands with one of its ducts indicated between the two arrowheads; r Ac, right acetabulum. Specimen fixed in 76° alcohol.

**Figure 5 animals-14-02748-f005:**
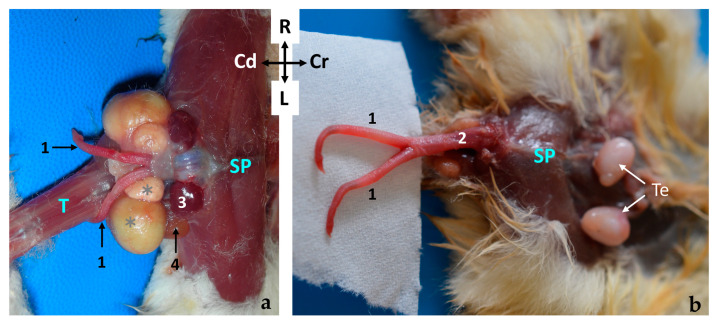
Ventral view of the caudal part of the body of a male sugar glider. Part of the penis is exteriorized through the cloacal orifice. Fresh specimen. (**a**) The free part of the penis is exteriorized through the cloacal orifice. (**b**) Displays both the body and the free part of the penis exteriorized through the cloacal orifice. 1, Bifurcated free part of the penis; 2, body of the penis; 3, left bulbospongiosus muscle; 4, left bulbourethral gland; ✱, left paracloacal glands; SP, symphysis pubica; T, tail; Te, testes. The orientation is the same in both images.

**Figure 6 animals-14-02748-f006:**
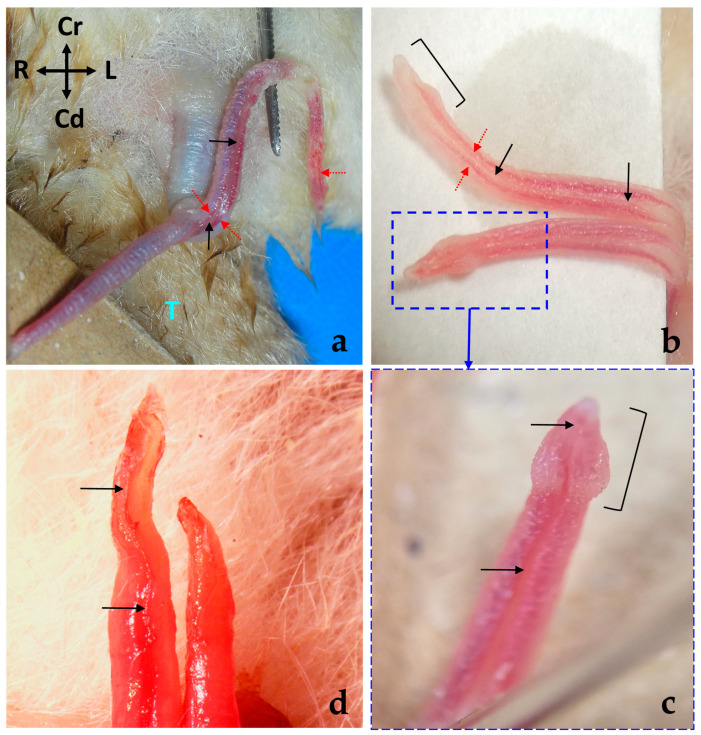
(**a**) Ventral view of the externalized penis of a male sugar glider in situ, exteriorized through the cloacal orifice. Fresh specimen. The left half of the free part has been displaced to observe its medial wall. (**b**) Medial wall of the bifurcated end of the penis, showing venous sinuses (red arrows). (**c**) Inset of (**b**) displaying the cone-shaped termination of the tip ( ] ) of the bifurcated penis, with a prominent widening at its base. Note the urethral groove (black arrows) continues up to the tip. (**d**) Tips of the bifurcated penis from a castrated male, showing the absence of the widened conical termination.

**Figure 7 animals-14-02748-f007:**
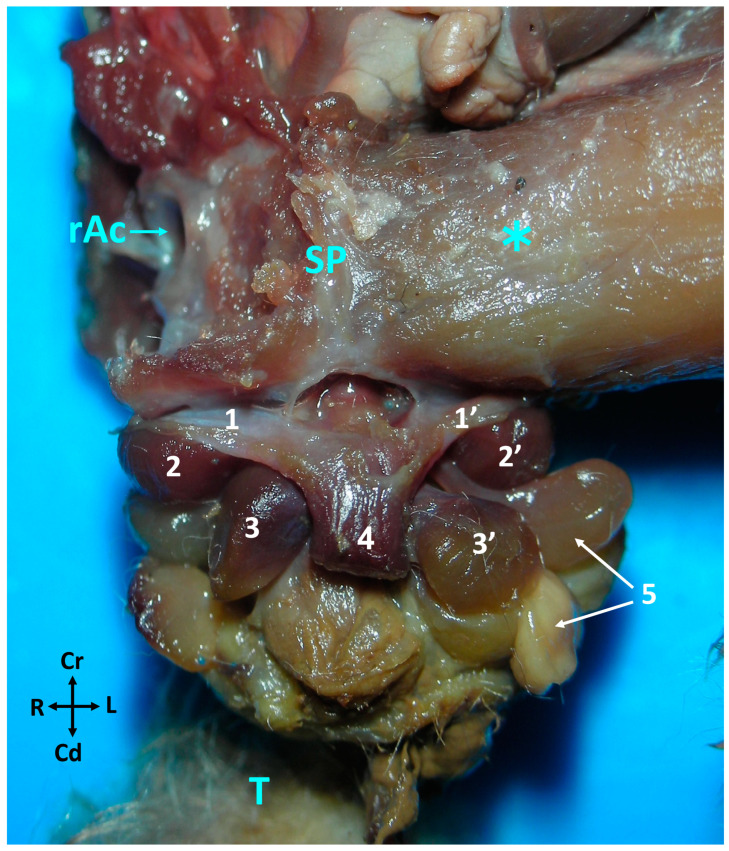
Ventral view of the caudal part of a male sugar glider to show the initial portion of the penis. The right leg has been removed. Fresh specimen. 1, 1′, Crura of the penis; 2, 2′, muscles ischiocavernosus; 3, 3′, muscles bulbospongiosus; 4, body of the penis; 5, left bulbourethral glands; SP, symphysis pubica; rAc, right acetabulum; T, tail; ✱ medial view of the left thigh.

**Figure 8 animals-14-02748-f008:**
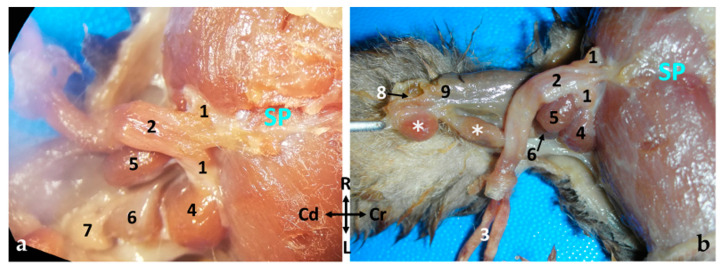
Examination of a fresh specimen. (**a**) Ventral view of a male sugar glider’s penis in situ. The anatomical structures on its left side have been dissected. (**b**) Ventral left view of a male sugar glider’s penis. The prepuce has been incised and the penis has been extracted from the preputial sac. 1, Crus penis; 2, corpus penis; 3, pars libera penis; 4, left ischiocavernosus muscle; 5, left bulbospongiosus muscle; 6, left bulbourethral gland I; 7, left bulbourethral gland II; 8, cloacal cavity; 9, rectum; ✱, paracloacal glands. SP, Symphysis pubica; both images have the same orientation.

**Figure 9 animals-14-02748-f009:**
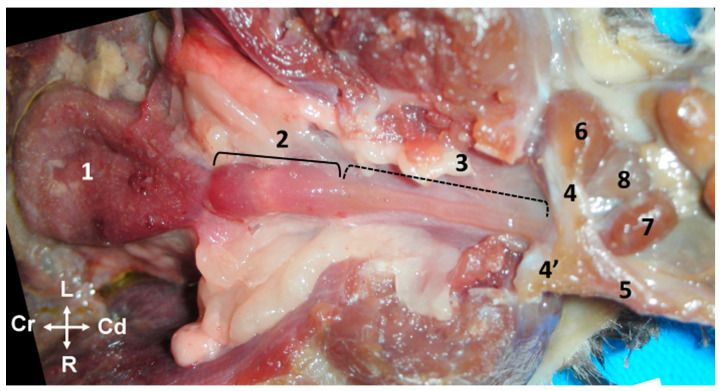
Ventral view of the prostate gland in situ. The floor of the pelvic cavity has been removed to observe the prostate and the urethra pelvina in the male sugar glider. Fresh specimen. 1, Bladder; 2, prostate; 3, urethra pelvina; 4, 4′, *Crura penis*; 5, *Corpus penis*; 6, left ischiocavernosus muscle; 7, left bulbospongiosus muscle; 8, left bulbourethral gland.

**Figure 10 animals-14-02748-f010:**
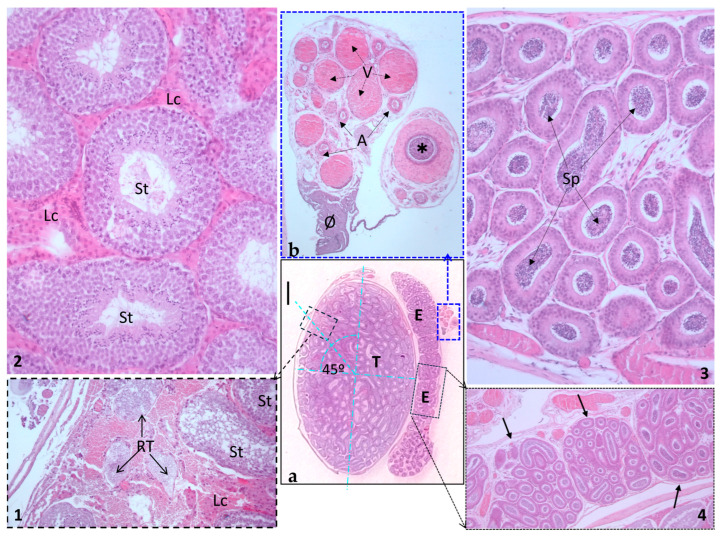
Testicle and epididymis. (**a**) A low-magnification section showing the testicle (T) and epididymis (E). Scale bar represents 1 mm. (1) An inset of (**a**) displays the rete testis, positioned at a 45º angle from the perpendicular axes (as depicted in (**a**)), from which the efferent duct transports spermatozoa to the epididymis. (2) Seminiferous tubules (St) teeming with numerous interstitial or Leydig cells (Lc). Note the distinct associations within the seminiferous tubules, exhibiting various stages of sperm cell development. (3) Histological section of the epididymis, revealing numerous cross-sections of the same epididymal duct filled with spermatozoa (Sp). (4) The black inset displays the characteristic lobuli epididymidis (black arrows) at a low magnification (objective 4×). (**b**) The blue inset in panel (**a**) provides a closer look at the spermatic cord and its components: ✱, deferent duct surrounded by a thick circular muscular layer; A, sections of the testicular artery; V, veins of the plexus pampiniformis; Ø, cremaster muscle (distal portion).

**Figure 11 animals-14-02748-f011:**
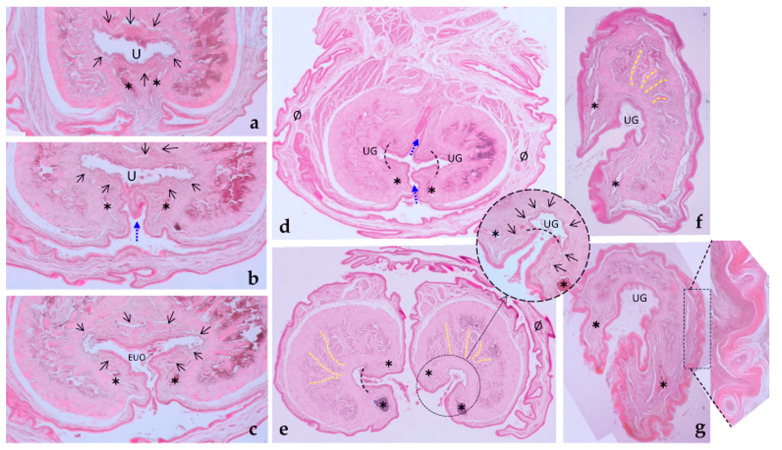
Penile histology. (**a**) The penile urethra is surrounded by the corpus spongiosum (black arrows) and the corpus cavernosum. (**b**) A ventral groove (blue arrow) deepens and almost reaches the penile urethra, just before the external urethral orifice. (**c**) The ventral groove reaches the urethra, which opens to the exterior, forming the external urethral orifice (EUO). (**d**) The groove continues dorsally (blue arrows) to form the bifurcated tip of the penis, and the continuation of the urethra creates two urethral grooves (UG) in a medial position of each part. Note the presence of two venous sinuses (✱) flanking the groove walls. The penis is protected by the mucosa of the prepuce (Ø). (**e**) The bifurcated penis is now separated from the prepuce, and each half is independent. The inset shows the urethral groove (UG), with urothelium at the groove’s bottom (the limits are indicated by the dotted curved line) and cutaneous mucosa forming the groove walls. This entire structure is surrounded by the corpus spongiosum (black arrows) and two venous sinuses on either side of the groove (one dorsal and one ventral (✱)). In the corpus cavernosum, yellow lines indicate dense connective tissue septa with a radial arrangement between blood-filled spaces. (**f**) The termination of the bifurcated penile tip, at the beginning of the conical formation. The corpus cavernosum is clearly visible, suggesting the absence of a glans. (**g**) An oblique section of the penile termination reveals cornified spines on the surface (inset). In both (**f**,**g**), the corpus cavernosum is evident, further supporting the absence of a glans.

**Figure 12 animals-14-02748-f012:**
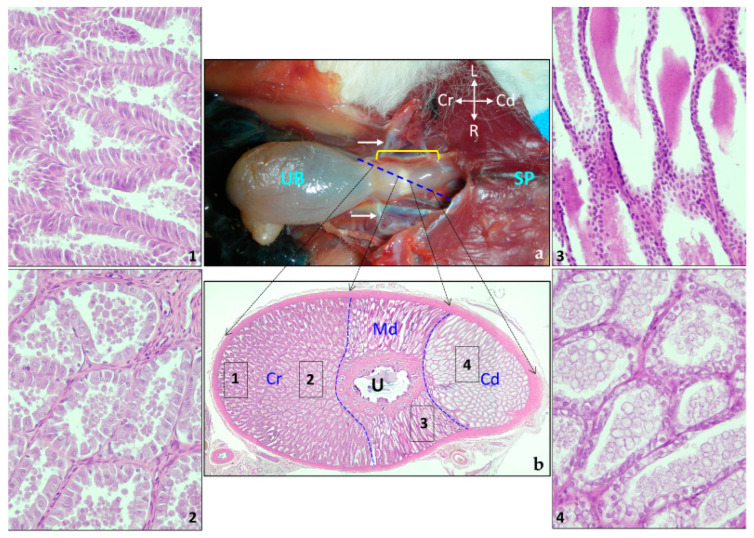
Prostate gland. (**a**) Macroscopic ventral view of the prostate gland in situ. The three parts of the prostate (yellow bracket) can be distinguished. White arrows, deferent ducts; SP, symphysis pubica; UB, urinary bladder. (**b**) Oblique histologic section of the prostate at the level of the blue dotted line in (**a**). The central image is a composition of three images made with the 4× objective and the numbers represent different areas of the cranial portion (Cr), middle (Md), and caudal part (Cd) easily distinguishable by their morphology. Photographs 1–4 represent the glandular tissue from the areas indicated in (**b**) to display their glandular epithelium and the secretion appearance (made with the 40× objective). U, Urethra.

**Figure 13 animals-14-02748-f013:**
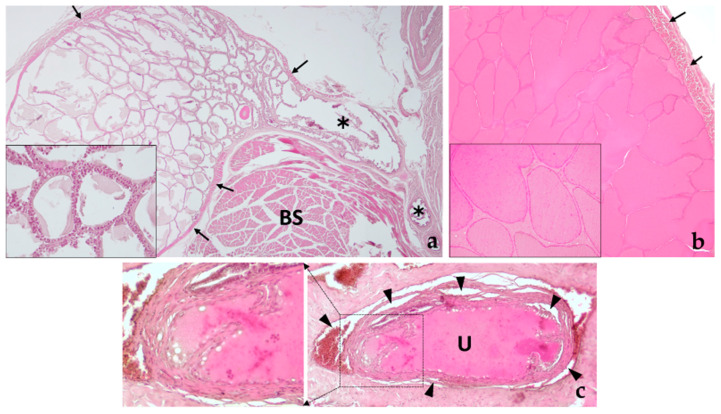
Histology of the bulbourethral glands. (**a**) Image showing bulbourethral gland I at low magnification to observe its tubulo-alveolar morphology and its secretion duct (✱). The entire gland is externally lined by striated muscle (black arrows) arranged in two perpendicular layers. The inset shows at higher magnification (40× objective) the morphology of the glandular epithelium made up of a single layer of cubic cells. BS, bulbospongiosus muscle. (**b**) Bulbourethral gland II, also of tubulo-alveolar nature, filled with acidophilic secretion. Two layers of striated muscle also appear surrounding the gland capsule. The inset shows at higher magnification (40× objective) the appearance of low cubic secretory cells arranged forming a single layer of cells lining the alveoli. The secretion, strongly acidophilic, presents some granules of more intense staining. (**c**) Section of the penile urethra showing the point of discharge of the ducts of the bulbourethral glands pouring their secretion into the lumen of the urethra. The inset shows in greater detail the discharge of the left duct. Note the presence of the spongy body surrounding the urethra (black arrowheads).

## Data Availability

Data sharing is not applicable to this article as all data associated is available in the text.

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
