# Peer review of "Anatomy of the Male Reproductive System of Sugar Gliders (Petaurus breviceps)"

_animals, 2024, doi:10.3390/ani14182748_

Round 1

Reviewer 1 Report

Comments and Suggestions for Authors

Notes for review, Anatomy of the male reproductive system of sugar gliders

Line 66 please consider if sexual dimorphism in this species is really subtle, as well as the differences you describe, the males weigh 20% more than females.

Line 109, “They are in the caudal part” the testes are not in the abdomen. Consider ‘ventral to’ or ‘beneath’ the caudal abdomen.

Line 179, Definitions of “perineal” vary, but most describe an area cranio-ventral to the anus extending to, but not including the scrotum or vulva. This definition is difficult to conceptualise in a marsupial. Please consider describing the location of the flaccid or retracted penis as “on the floor of the cloaca and pelvic canal”

Line 191, “At the botton of the sac” please consider ‘where the mucosa of the preputial sac reflects, it attaches to the penis’ or ‘at the deepest part of the preputial sac, the mucosa reflects onto the penis’

Line 217, “is situated at the base of the forked end  please consider ‘is situated in the bifurcation of the penis’

Line 232 “these columns” this is the first mention of these substructures within the body of the penis, but the concept of “columns” isn’t adequately described till line 371. Consider some more explanation here where the word is first used.

line 253, In figure 7, it is the right leg which has been removed.

Line 272, “the conical tip of the penis, unless in the castrated specimen.” Please consider “‘except’ in the castrated specimen”

Line 335 & Figure 10b, 

Comments on the Quality of English Language

A couple of suggestions included in file

Author Response

 Please, see attachment.

Reviewer 2 Report

Comments and Suggestions for Authors

Thank you for doing this. It is needed for sugar glider practitioners as they do not usually understand the comparisons with placental mammals. Please see all the comments that I put in the uploaded file.  There is one reference you need to include - the Meredith Smith chapter in Possums and Gliders that has good anatomical drawings, and starts comparisons. If you do not have this, if you let the editors know, I forward the pdf to you. 

Reviewer 3 Report

Comments and Suggestions for Authors

1. Sumary
The manuscript aims to provide a detailed anatomical description of the male reproductive system in sugar gliders, a species of marsupial that has gained popularity as a pet. The study's main contributions include a comprehensive examination of both macroscopic and microscopic anatomical features, offering valuable insights for veterinary clinicians. The strengths of the paper lie in its detailed anatomical descriptions and the potential to enhance clinical practices related to the care of sugar gliders.

2. General comments

The article is clear, well-structured, well-written, and technically sound. The images effectively illustrate the anatomy described. The discussion comprehensively covers other marsupial species in a well-elaborated, clear, and robust manner.

Due to the excellent quality of the manuscript, this Reviewer did not find any points worthy of comment to add to this review and congratulates the authors on their outstanding work. Understanding reproductive anatomy is one of the first steps in developing reproductive biotechnologies for species conservation, especially from the perspective of the One Conservation concept. Congratulations on your efforts to generate knowledge that can also be applied to conservation.

In an attempt to contribute in some way, and by no means holding back the acceptance of this manuscript for publication, this Reviewer offers the following perspective:

a) It is often possible to use species of the same genus as experimental models for threatened species.

b) Petaurus breviceps is classified as Least Concern regarding its global extinction risk by the IUCN.

c) However, there are three other Petaurus species in more vulnerable situations: P. australis (Near Threatened), P. gracilis (Endangered), and P. abidi (Critically Endangered).

Therefore, it is worthwhile in future works to include a brief discussion on the importance of knowledge, as described in this article, of a more abundant species as an experimental model for more threatened species. This is particularly relevant if one of the objectives is to advance reproductive biotechnologies for the conservation of these species. The suggestion is to explore the One Conservation concept, which the authors can learn about here: https://doi.org/10.1590/1984-3143-AR2021-0024.

3. Specific comments

Should some words be in italics, such as in the lines below, and other Latin terms throughout the text?

Line 109 - (testis)

Line 178 - (penis)

Line 263 - urethra masculina

Line 264 - ostium urethrae externum
